# Improved Pseudo Data for Machine Translation Quality Estimation with Constrained Beam Search

**Xiang Geng**[1*]    **Yu Zhang**[1*]    **Zhejian Lai**[1]    **Shuaijie She**[1]    **Wei Zou**[1]

**Shimin Tao**[2]    **Hao Yang**[2]    **Jiajun Chen**[1]    **Shujian Huang**[1] [†]

[1] National Key Laboratory for Novel Software Technology, Nanjing University, Nanjing, China

[2] Huawei Translation Services Center, Beijing, China

{gx,zhangy,laizj,shesj,zouw}@smail.nju.edu.cn

{taoshimin,yanghao30}@huawei.com,{chenjj,huangsj}@nju.edu.cn

## Abstract

Machine translation (MT) quality estimation (QE) is a crucial task to estimate the quality of MT outputs when reference translations are unavailable. Many studies focus on generating pseudo data using large parallel corpus and achieve remarkable success in the supervised setting. However, pseudo data solutions are less satisfying in unsupervised scenarios because the pseudo labels are inaccurate or the pseudo translations differ from the real ones. To address these problems, we propose to generate pseudo data using the MT model with constrained beam search (CBSQE). CBSQE preserves the reference parts with high MT probabilities as correct translations, while the rest parts as the wrong ones for MT generation. Therefore, CBSQE can reduce the false negative labels caused by synonyms. Overall, beam search will prefer a more real hypothesis with a higher MT generation likelihood. Extensive experiments demonstrate that CBSQE outperforms strong baselines in both supervised and unsupervised settings. Analyses further show the superiority of CBSQE. The code is available at https://github.com/NJUNLP/njuqe.

## 1  Introduction

With the rapid development of machine translation (MT), evaluating the quality of MT outputs becomes increasingly essential. Common MT metrics, such as BLEU (Papineni et al., 2002) or TER (Snover et al., 2006), rely on reference translations which are unavailable in most applications of MT systems. Quality estimation (QE) is the task of automatically assessing the quality of machine translations at run-time without relying on references (Specia et al., 2018). QE plays an important role in real-world scenarios. For example, QE improves post-editing (PE) workflows by selecting high-quality MT outputs and indicating incorrect

---

*Equal Contribution
†Corresponding Author

| | |
|---|---|
| **Source** | the Symbol screener tool applies opacity to symbol instances . |
| **MT** | mit dem *Symbol-aufzeichner-Werkzeug* können Sie die Deckkraft *auf* Symbol-instanzen *anwenden* . ‖ HTER = 0.2727 |
| **PE** | mit dem Symbolaufzeichnungs-Werkzeug können Sie die Deckkraft für Symbol-instanzen festlegen . |

Table 1: An example from the WMT19 English-German (EN-DE) QE dataset. For word-level tags, the token with translation error is labeled "BAD" (indicated in italic font with red color); otherwise, it is labeled "OK". Sentence-level score HTER measures the whole effort of manually correcting the MT based on edit distance.

translation words (Specia, 2011); QE can guide decoding (Wang et al., 2020) and re-ranking process (Bhattacharyya et al., 2021) to improve the translation performance.

As shown in Table 1, sentence- and word-level quality labels can be derived by matching the hypothesis with post-edit references using TER tools. As noted by Snover et al. (2006), post-editing considers the semantic equivalence and preserves correct parts of MT outputs. Therefore, TER tools will acknowledge the semantically correct translations simply by exact matching. Although post-editing improves quality labels, it remarkably increases annotation costs.

Therefore, many studies turn to generate pseudo QE data (pseudo MT outputs and pseudo QE labels) using the large parallel corpus. Furthermore, these pseudo data are used to pre-train a supervised model or train an unsupervised model. DirectQE (Cui et al., 2021) uses a masked language model conditioned on source sentence as the translation language model (TLM) to generate pseudo translations. Given a parallel pair, Cui et al. (2021) randomly mask reference tokens and replace them with ones sampled from the TLM generation distribution. The replaced tokens are annotated as errors,

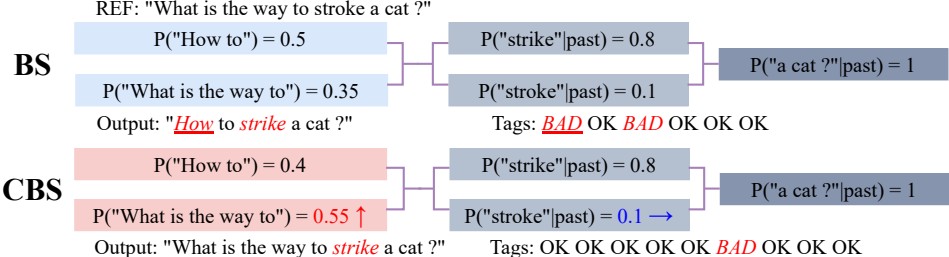

Figure 1: A toy example of CBSQE in English with beam size 2. "BAD" tags are marked in italic font with red color and will be further underlined if their labels are wrong. CBSQE increases the generation probability $P$("What is the way to") of the synonym while keeping $P$("strike"|past) of the translation error.

and the sentence score is calculated as the ratio of replacement. BSQE (Tuan et al., 2021) directly generates pseudo translations using a neural machine translation (NMT) with the beam search algorithm (BS) and obtains labels by matching these translations with corresponding references using TER tools.

Although pseudo data solutions achieve remarkable performance in the supervised setting (Cui et al., 2021), they are less satisfying in unsupervised scenarios (Zheng et al., 2021) due to generation noise. E.g., DirectQE yields repetition or incomplete translation which is different from the real ones due to its non-autoregressive generation (Wang et al., 2019). Furthermore, its negative sampling strategy sacrifices the quality of pseudo translation for accurate pseudo labels. On the other hand, BSQE suffers from inaccurate pseudo labels. As shown at the top of Figure 1, the default beam search yields the synonyms of the reference sentence. However, it is difficult for existing reference-based evaluation methods to match these synonyms correctly, especially when the sentence structure changes too much.

To improve the quality of pseudo data, we propose to generate pseudo translations using an MT model with constrained beam search (CBSQE). CBSQE corrupts translations with reference parts by adjusting the generation probability. CBSQE improves the probability of references when the probability of any reference token is above a threshold so that the less fluent hypothesis will not be mistakenly rewarded. Besides, we design an adjustment function that assigns larger improvements to closer reference tokens. This allows us to preserve the main structure of references thereby obtaining accurate labels using TER tools. We show an example at the bottom of Figure 1.

Our main contributions can be summarized as follows:

- We propose a novel and general pseudo data generation method that reconciles the pseudo label with pseudo translation quality.

- The proposed method is viable in evaluations based on pseudo data and superior in both supervised and unsupervised settings.

- The analysis reveals that the designed threshold strategy and adjustment function matter. Besides, CBSQE is efficient and open source.

## 2 Preliminary

### 2.1 Machine Translation

Machine translation (MT) models a conditional generation of a target sequence $\mathbf{y} = \{y_1, y_2, \ldots, y_n\}$ given a source sentence $\mathbf{x}$. The generation can be described in an autoregressive style as $P(\mathbf{y}|\mathbf{x}) = \prod_{t=1}^{n} P(y_t|\mathbf{y}_{<t}, \mathbf{x}; \theta)$, where $\mathbf{y}_{<t}$ denote the translation history and $\theta$ is the parameter set. The translation is acquired by solving $\hat{\mathbf{y}} = \arg\max_{\mathbf{y}} P(\mathbf{y}|\mathbf{x}; \theta)$ via beam search. The beam search can be further guided by additional penalties for constrained generation (Hokamp and Liu, 2017; Hu et al., 2019). Although beam search yields fluent translations, its diversity of utterance is lesser than the actual parallel corpus (Zhou et al., 2019). Thus, the MT outputs are often evaluated by references in distinct utterances (Chatterjee et al., 2019).

In many applications, we have access to the target MT model that needs to be evaluated. Following this trend, WMT QE shared tasks have adopted the white-box setting in recent years (Specia et al., 2021). In the black-box setting, we can obtain a competitive multilingual MT model (Tang et al., 2020) for low-resource languages.

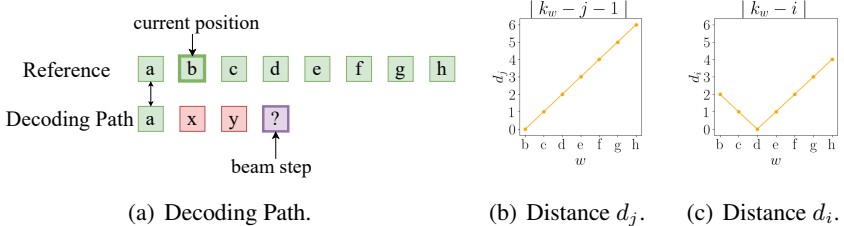

(a) Decoding Path.    (b) Distance $d_j$.    (c) Distance $d_i$.

Figure 2: Illustration for adjusting generation probabilities.

## 2.2 Quality Estimation for Machine Translation

QE assesses the quality of translation without access to references. The evaluation is conducted in different grains: word-, sentence- or document-level. Besides, there are several types of supervision for QE as noted in (Belouadi and Eger, 2022): (1) the supervised setting with both QE and parallel data accessible; (2) the unsupervised setting without QE data, but parallel data accessible; (3) the fully unsupervised setting without QE and parallel data. In this paper, we focus on word- and sentence-level QE in supervised and unsupervised settings which cover plenty of language directions.

Given a source language $\mathbf{x}$ and a target language translation $\hat{\mathbf{y}} = \{y_1, y_2, \ldots, y_n\}$, the word-level labeling is a sequence of tags $\mathbf{g} = \{g_1, g_2, \ldots, g_n\}$, where $g_i =$"OK" represents the corresponding $y_i$ is correct while "BAD" to be further post-edited. The sentence-level HTER score $h$ indicates the overall edit effort of correcting arbitrary $\hat{\mathbf{y}}$ into its post-edited reference:

$$h = \frac{\text{number of edits}}{\text{post-editing sentence length}}. \quad (1)$$

These labels are produced via TER tools by taking the translation and PE sequences as inputs.

## 3 Method

We first introduce the constrained beam search algorithm, then the training and inference procedures of the QE model.

### 3.1 Constrained Beam Search

As aforementioned in Section 2.1, due to the difference between translations and references, many correct translation words will be mistakenly labeled as negative by TER tools. Synonym annotation requires fine-grained reference-based evaluation, which is yet another challenging task like QE since

both structure and semantic equivalence are involved. As shown in Algorithm 1, we propose a constrained beam search algorithm (CBS) to solve this problem at the decoding stage instead of the annotation stage. Inspired by targeted human annotation (Snover et al., 2006), we constrain the decoding algorithm to preserve parts of references that the MT model tends to generate. To this end, it is essential to decide when to preserve references and which reference tokens should be preserved.

**Threshold Strategy.** We expect to generate fluent sentences with higher NMT probabilities as well as reflect real translation errors. Thus, CBSQE preserves reference tokens when the probability of any reference token is above a threshold $\epsilon$ (line 9 in Algorithm 1).

**Adjustment Function.** On the other hand, we want to preserve the main structure of references so that we can obtain accurate labels using TER tools. Therefore, we design an adjustment function that assigns larger improvements to closer reference tokens when adjusting generation probabilities. Consider the example in Figure 2(a), we need to define which are closer tokens. There are two extremes: (i) "x" and "y" are the insertion errors, and the rest decoding parts are aligned with reference tokens $\mathbf{r}_{>j} =$ "b c d e f g h", where $j$ denotes the position of the last reference token being decoded. The distance from $w$ to the anchor $\mathbf{r}_{j+1} =$"b" is $d_j = |k_w - j - 1|$ (Figure 2(b)), where $k_w$ is the position of the nearest $w \in r_{>j}$ from left; (ii) "x" and "y" are the substitution errors, and the rest decoding parts are aligned with reference words $\mathbf{r}_{>i} =$ "d e f g h", where $i$ denotes the beam step. We have $d_i = |k_w - i|$ (Figure 2(c)). We assume that these two extremes occur with equal probability, thus we define the adjustment function as follows:

$$A(w) = \begin{cases} \frac{1}{(d_j + d_i)^2} & \text{if } w \in \mathbf{r}_{>j} \\ -\text{INF} & \text{if } w \notin \mathbf{r}_{>j} \end{cases}. \quad (2)$$

**Algorithm 1** Constrained Beam Search.

---

**Input**: Parallel pair $(\mathbf{x}, \mathbf{r})$, beam size $k$, translation model $\theta$ with vocabulary $\mathbf{w}$, temperature $\tau$, threshold $\epsilon$, combined weight $\lambda \in [0, 1]$.

**Output**: Pseudo translation $\hat{\mathbf{y}}$.

1:   $\mathcal{H}_{\text{cur}} \longleftarrow \{(\text{BOS}, 0, 0)\}$, $i = 1$. # Initialization.
2:   **repeat**
3:      $\mathcal{H}_{\text{next}} \longleftarrow \emptyset$.
4:      **for all** $\{(\mathbf{y}, p, j)\} \in \mathcal{H}_{\text{cur}}$ **do**
5:        **if** $\mathbf{y}_{|\mathbf{y}|} ==$ EOS **then**
6:          $\mathcal{H}_{\text{next}} \longleftarrow \mathcal{H}_{\text{next}} \cup \{(\mathbf{y}, p, j)\}$.
7:        **else**
8:          $\mathbf{p} = P(\mathbf{w}|\mathbf{y}, \mathbf{x}; \theta)$.
9:          **if** $\exists w \in \mathbf{r}_{>j}, \mathbf{p}_w > \epsilon$. **then**
10:            $\mathbf{q} = \text{softmax}(\frac{A(\mathbf{w})}{\tau})$. # Normalize $A(\mathbf{w})$ with softmax function.
11:            $\mathbf{p} = \lambda \mathbf{q} + (1 - \lambda)\mathbf{p}$.
12:          **end if**
13:          $\mathcal{H}_{\text{next}} \longleftarrow \mathcal{H}_{\text{next}} \cup \bigcup_{w \in \mathbf{w}} (\mathbf{y} \cdot w, p \times \mathbf{p}_w, \max(k_w, j))$. # · denotes the concatenate operation.
14:        **end if**
15:      **end for**
16:      $\mathcal{H}_{\text{cur}} \longleftarrow \{(\mathbf{y}, p, j) \in \mathcal{H}_{\text{next}} : |\{(\mathbf{y}', p', j') \in \mathcal{H}_{\text{next}} : p' > p\}| < k\}$. # Select k-best candidates.
17:      $(\hat{\mathbf{y}}, \hat{p}, \hat{j}) = \arg\max_{(\mathbf{y}, p, j) \in \mathcal{H}_{\text{cur}}} p$, $i = i + 1$.
18: **until** $\hat{\mathbf{y}}_{|\hat{\mathbf{y}}|} =$ EOS

---

We calculate the adjustment over vocabulary $\mathbf{w}$ and normalize the results using the softmax function with temperature $\tau$ (line 10 in Algorithm 1). Finally, we combine the increment $\mathbf{q}$ with original probabilities $\mathbf{p}$ by weight $\lambda$:

$$\mathbf{p} = \lambda \mathbf{q} + (1 - \lambda)\mathbf{p}. \quad (3)$$

**Beam Search.** Since the generation procedure is autoregressive, incremental adjustments may relatively decrease the future decoding probabilities. Therefore, we adopt beam search to find the best candidate to meet our constraints. That is, we keep a set of active candidates $\mathcal{H}_{\text{cur}}$. In each iteration, we expand all incomplete candidates (without EOS) as aforementioned and place them into $\mathcal{H}_{\text{next}}$ (line 4-15 in Algorithm 1). Then we collect the best $k$ candidates in $\mathcal{H}_{\text{next}}$ according to the adjusted probabilities as the active set $\mathcal{H}_{\text{cur}}$ in the next iteration (line 16 in Algorithm 1). We iteratively refine the best candidate in $\mathcal{H}_{\text{cur}}$ until the end of sequence (EOS).

Overall, CBSQE maintains the main structure of the reference, which ensures the labeling accuracy of TER tools. The mismatched parts compared to reference are hardly labeled false-negative since their generation probabilities are distinct compared to those of the correct parts.

### 3.2 Training and Inference

With the development of pre-trained language models (PLMs), multilingual PLMs such as mBERT (Devlin et al., 2019) and XLMR (Conneau et al., 2020) have been widely used for initializing QE models (Kim et al., 2019; Ranasinghe et al., 2020). In this work, we adopt the XLMR-large model as the base model for all pseudo data solutions. Given a (pseudo) QE sample $(\mathbf{x}, \hat{\mathbf{y}}, \mathbf{g}, h)$, we concatenate the source $\mathbf{x}$ and the translation $\hat{\mathbf{y}}$ as the input. We take the corresponding outputs from the last layer as representations of each sub-tokens. The word representations are the average across representations of all sub-tokens. Likewise, the sentence score representation averages all representations of the target sub-tokens. These representations are passed through linear layers to predict word tags and sentence scores, respectively.

We pre-train and fine-tune the QE model under the multi-task learning framework by summing up the objective of the sentence- and word-level tasks:

$$J_{\text{QE}} = \log P(h|\mathbf{x}, \hat{\mathbf{y}}; \theta) + \log P(\mathbf{g}|\mathbf{x}, \hat{\mathbf{y}}; \theta), \quad (4)$$

where sentence- and word-level tasks are regarded as regression and sequence labeling, respectively.

Fomicheva et al. (2020) and Zheng et al. (2021) quantify NMT or TLM model uncertainty using the

Monte Carlo dropout (Gal and Ghahramani, 2016). The obtained uncertainty features have achieved remarkable performance for unsupervised QE. This strategy can also be regarded as a self-ensemble technique (Laine and Aila, 2016). In this paper, we also explore Monte Carlo dropout for CBSQE. Specifically, we use dropout to perturb the QE model and predict the QE labels with the perturbed model several times. Then, we average all output scores as the final result for the sentence-level task and obtain word-level results by voting.

## 4 Experiments

### 4.1 Setup

**Datasets.** We conduct the experiments on the widely used benchmarks for QE from WMT[1] QE shared tasks: WMT19 English-German (EN-DE), WMT20/21 English-Chinese (EN-ZH). WMT19, 20, and 21 datasets contain 13K/1K/1K, 7K/1K/1K, and 8K/1K/1K QE samples for training, validation, and testing, respectively. In the main experiments of each language direction, we only use the same 500K parallel pairs for all QE methods. These datasets are all officially released by WMT organizers and have already been tokenized and true-cased.

**Baselines.** We compare the CBSQE with strong baselines, including the pseudo data method (DirectQE (Cui et al., 2021) and BSQE (Tuan et al., 2021)), the unsupervised method based on uncertainty features (SSQE (Zheng et al., 2021)). We also build a standard supervised baseline XLM-RQE by directly fine-tuning the XLMR using real QE data.

**Implementation Details.** We reproduce the SSQE using the codes released by Zheng et al. (2021)[2]. SSQE also employs a large pre-trained language model, mBERT-large. We follow the guide by Zheng et al. (2021) to set hyper-parameters and search for the best threshold for word-level tags using the validation set. We implement all pseudo data methods in NJUQE based on the open-source toolkit Fairseq (Ott et al., 2019). We follow the setting of (Cui et al., 2021) to train the TLM for all QE tasks. We train an NMT model for WMT19 EN-DE, which is the Transformer-base setting (Vaswani et al., 2017). Since WMT20/21

[1]https://www.statmt.org/
[2]https://github.com/THUNLP-MT/SelfSupervisedQE

EN-ZH tasks adopt the glass-box setting, we directly use the target NMT model provided by WMT. We use the TER tool called TERCOM[3] to annotate the pseudo translations generated by NMT or CBSQE. Our experiments are conducted on NVIDIA 3090-Ti/V100 GPUs. We provide more implementation details in Appendix A.

**Evaluation Metrics.** Following WMT QE shared tasks, the sentence-level task will be evaluated using Pearson's correlation coefficient. For the word-level task, WMT19 used F1-MULT as the primary metric, while WMT20/21 conducted the Matthews correlation coefficient (MCC) as the primary metric. F1-MULT is denoted as the multiplication of F1-scores for the "OK" and "BAD" words. We provide the results of MAE and RMSE metrics in Appendix A for reference.

### 4.2 Main Results

Table 2 shows the main results for baselines and CBSQE on different QE datasets. Monte Carlo dropout is essential for the SSQE and can slightly improve the performance of CBSQE. We only use this strategy for a fair comparison between CBSQE and SSQE because it significantly increases the inference times. For word-level tasks, CBSQE mainly improves the F1-BAD score, which implies that CBSQE could reduce false-negative labels.

For all unsupervised tasks, CBSQE significantly outperforms strong baselines (significance test can be found in Appendix C). Specifically, CBSQE (w/ dropout) increases the Pearson coefficient by **+3.18** and the F1-MULT by **+0.93** on the WMT19 EN-DE dataset. For WMT20/21 EN-ZH tasks, CBSQE (w/ dropout) increases the Pearson coefficient by **+4.69/+7.42** and the MCC by **+4.25/+2.71** on WMT20 and WMT21 tasks, respectively. We achieve more improvements on EN-ZH than EN-DE. We assume there are more synonyms in Chinese, while CBSQE successfully reduces the gap between pseudo translations and references. Furthermore, we improve the performance in unsupervised settings towards that of XLMRQE in supervised settings.

For all supervised scenarios, CBSQE slightly improves the performance across the board compared to alternative pseudo data methods. Fine-tuning on small QE datasets involves inductive bias so that it reduces the benefits of improved pseudo data.

[3]http://www.cs.umd.edu/~snover/tercom/

| Dataset | Settings | Method | Sent-level Test | Word-level Test | | | |
|---|---|---|---|---|---|---|---|
| | | | Pearson↑ | MCC↑ | F1-MULT↑ | F1-OK↑ | F1-BAD↑ |
| WMT19 EN-DE | Unsupervised | DirectQE | 42.52 | 33.88 | 32.74 | **93.02** | 35.19 |
| | | BSQE | 46.30 | 31.98 | 36.59 | 86.15 | 42.47 |
| | | SSQE | 40.57 | 30.58 | 35.95 | 90.88 | 39.56 |
| | | SSQE (w/ dropout) | 45.04 | 32.92 | 38.41 | 90.45 | 42.46 |
| | | **CBSQE** | 48.47 | 33.90 | 38.71 | 87.56 | 44.20 |
| | | **CBSQE (w/ dropout)** | **49.48** | **34.45** | **39.34** | 88.11 | **44.65** |
| | Supervised | XLMRQE | 52.21 | 38.54 | 42.70 | 91.91 | 46.46 |
| | | DirectQE | 56.26 | 42.21 | 45.88 | 92.46 | 49.62 |
| | | BSQE | 56.45 | 41.87 | 45.73 | 92.19 | 49.60 |
| | | **CBSQE** | **56.65** | **42.69** | **46.25** | **92.66** | **49.92** |
| WMT20 EN-ZH | Unsupervised | DirectQE | 45.15 | 32.32 | 43.47 | 66.12 | 65.74 |
| | | BSQE | 54.99 | 36.12 | 46.25 | 65.86 | 70.23 |
| | | SSQE | 40.15 | 30.70 | 42.65 | 63.08 | 67.61 |
| | | SSQE (w/ dropout) | 43.82 | 33.50 | 44.52 | 64.98 | 68.52 |
| | | **CBSQE** | 59.09 | 39.72 | 48.71 | 67.61 | **72.05** |
| | | **CBSQE (w/ dropout)** | **59.68** | **40.37** | **49.23** | **68.55** | 71.82 |
| | Supervised | XLMRQE | 60.55 | 45.78 | 52.10 | 73.42 | 70.96 |
| | | DirectQE | 63.59 | 49.02 | 54.88 | 74.65 | 73.52 |
| | | BSQE | 64.71 | 48.85 | 54.84 | 74.50 | 73.61 |
| | | **CBSQE** | **65.42** | **49.74** | **55.67** | **74.74** | **74.49** |
| WMT21 EN-ZH | Unsupervised | DirectQE | 20.35 | 15.72 | 27.03 | **83.25** | 32.47 |
| | | BSQE | 21.22 | 17.49 | 28.74 | 77.38 | 37.14 |
| | | SSQE | 21.39 | 20.35 | 30.92 | 80.10 | 38.61 |
| | | SSQE (w/ dropout) | 21.40 | 19.79 | 30.12 | 77.49 | 38.87 |
| | | **CBSQE** | 27.41 | 22.53 | 32.06 | 78.83 | **40.67** |
| | | **CBSQE (w/ dropout)** | **28.82** | **23.06** | **32.92** | 82.14 | 40.08 |
| | Supervised | XLMRQE | 30.38 | 28.01 | 35.63 | 80.49 | 44.27 |
| | | DirectQE | 31.99 | 30.15 | 36.92 | 80.02 | 46.13 |
| | | BSQE | 31.88 | 29.45 | 36.38 | 79.69 | 45.65 |
| | | **CBSQE** | **32.15** | **30.97** | **37.63** | **80.58** | **46.70** |

Table 2: Main results for baselines and CBSQE on different QE test sets.

| Pseudo Data | $P(y|x;\theta)$ ↑ | Acc-All ↑ | Acc-BAD ↑ |
|---|---|---|---|
| DirectQE | 0.0613 | **94.99%** | **88.11%** |
| BSQE | **0.7018** | 63.00% | 35.71% |
| CBSQE | 0.6905 | 88.37% | 58.76% |

Table 3: Average probability of the target NMT model and word tags accuracy for different pseudo data methods on EN-ZH direction.

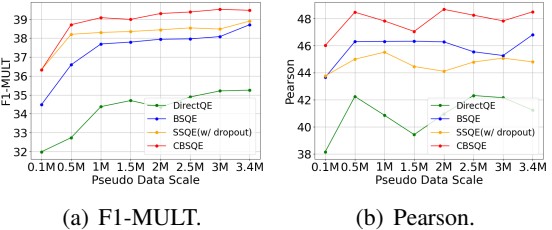

(a) F1-MULT.

(b) Pearson.

Figure 3: (a) F1-MULT or (b) Pearson score of different methods on the WMT19 EN-DE QE test set using different amounts of parallel pairs.

## 4.3 Amount of Pseudo Data

We investigate how data size affects the performance of unsupervised QE methods. We conduct further experiments using different amounts of parallel pairs for different QE methods on the WMT19 EN-DE dataset. As shown in Figure 3, CBSQE consistently outperforms baselines with different data sizes. For the word-level task, pseudo data methods achieve remarkable improvements with the growth of data size (Figure 3(a)). However, using over 500K data does not significantly improve the sentence-level results for all methods (Figure 3(b)). We assume that the sentence-level scores follow obvious preferences such as fluency (Sun et al., 2020), causing a limited contribution of increased data size.

## 5 Analysis

In unsupervised settings, we analyze the quality of CBSQE pseudo data, factors contributing to the improvement, and the efficiency of CBSQE.

| Source | This is an important development that deserves to be widely adopted . |
|---|---|
| **Reference** | 这是 一个 值得 广泛 采纳 的 重要 发展 。 |
| **DirectQE** | 这是 一个 值得 *值得 说明* 的 重要 *企业* 。 |
| **& Its Back** | This is an important *enterprise* that deserves *deserves* to be *explained* . |
| **BSQE** | 这是 一项 重要 的 发展 ， 应该 得到 广泛 采纳 。 |
| **& Its Back** | This is *an important development that deserves to be widely adopted* . |
| **CBSQE** | 这是 一个 值得 广泛 采纳 的 重要 *事态* 发展。 |
| **& Its Back** | This is an important development *of the situation* that deserves to be widely adopted . |

Table 4: Pseudo QE data for EN-ZH QE task. The tokens with "BAD" tags are marked in italic font with red color and will be further marked with the underline if their labels are wrong.

## 5.1 Improved Pseudo Data

**Quantitative analysis.** We evaluate the quality of pseudo data from two aspects: (1) the quality of pseudo translations; (2) the accuracy of the pseudo labels. To answer the first issue, we check the target NMT's generation likelihoods of the pseudo translations by averaging the NMT probabilities of all 3.4M pseudo translations generated by different methods. To answer the second issue, we randomly sample 40 pseudo samples for each method and manually check the accuracy of word tags. The results are summarized in Table 3. The negative sampling strategy improves the pseudo label accuracy of DirectQE. However, DirectQE achieves relatively low NMT probabilities because of the non-autoregressive TLM and negative sampling strategy. CBSQE obtains improved word tag accuracy than BSQE by reducing the false negative. Using threshold $\epsilon$ and beam search, CBSQE achieves a slightly lower NMT probability than BSQE though CBSQE constrains the MT model to preserve the major reference parts.

**Qualitative analysis.** Table 4 shows the pseudo data generated by different methods for EN-ZH tasks. As aforementioned, DirectQE generates irrelevant pseudo translation with accurate labels; BSQE generates translations with noisy labels because of synonyms; CBSQE successfully generates proper translation errors with accurate labels.

## 5.2 Ablation Studies

We further conduct ablation studies with different CBSQE variants on the WMT19 EN-DE dataset.

**Effect of threshold $\epsilon$.** To measure the effect of threshold $\epsilon$, we perform CBSQE with $\epsilon = 0$ so that CBSQE will increase reference tokens, whatever their probabilities. Table 5 shows that threshold matters for CBSQE.

| Method | Pearson↑ | F1-MULT↑ |
|---|---|---|
| CBSQE w/o threshold | 46.80 | 34.03 |
| CBSQE w/o adjustment function | 42.62 | 35.48 |
| CBSQE | **48.47** | **38.71** |

Table 5: Ablation studies on the WMT19 EN-DE test set.

| Method | Generation Time (ms) ↓ | Speed ↑ |
|---|---|---|
| DirectQE | 17.45 | 4.3× |
| BSQE | 75.06 | 1× |
| CBSQE | 100.07 | 0.75× |

Table 6: Average generation time per sample of different pseudo data methods on a single 3090Ti GPU.

**Effect of adjustment function $D$.** The adjustment function $D$ assigns larger probabilities to reference words that are closer to current position $j$ or beam step $i$. Without adjustment function $D$, CBSQE assigns the same adjustment to each reference word and causes performance degradation (Table 5).

## 5.3 Efficiency

**Generation timelapse.** We record the generation timelapse of different methods for 125K samples. As shown in Table 6, DirectQE generates pseudo data 4.3× faster than BSQE due to the non-autoregressive decoding. Although CBSQE introduces complex constraints, the computational complexity does not increase too much (only 0.75× of the original timelapse).

**Convergence efficiency.** Better pseudo data could speed up training convergence. To compare the convergence of different methods, we plot the learning curve on the WMT19 EN-DE validation set in Figure 4. For both sentence- and word-level tasks, CBSQE converges faster than DirectQE and BSQE, achieving the highest performance.

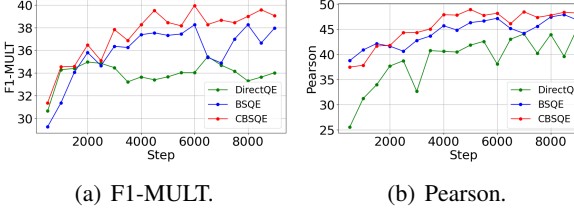

(a) F1-MULT.     (b) Pearson.

Figure 4: Training step vs. (a) F1-MULT or (b) Pearson score of different methods on the WMT19 EN-DE QE validation set.

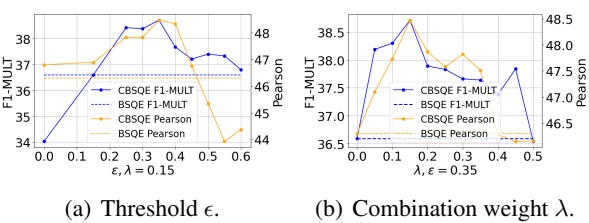

(a) Threshold $\epsilon$.   (b) Combination weight $\lambda$.

Figure 5: Performance of CBSQE on the WMT19 EN-DE QE test set using different (a) threshold $\epsilon$ and (b) combination weight $\lambda$.

**Tuning hyper-parameters.** CBSQE involves three hyper-parameters. Specifically, the temperature $\tau$ determines how much probability is assigned to each position. We simply set $\tau = 5$ for all experiments. The threshold $\epsilon$ decides when to preserve reference words. The combination weight $\lambda$ decides the BAD ratio of pseudo translations. If we set $\lambda = 1$, the generated pseudo translations will be the same as references, and the BAD ratio equals 0; if we set $\lambda = 0$, we obtain real MTs, and the BAD ratio equals TER. As shown in Figure 5, CBSQE outperforms baselines under the broad range of $\epsilon$ and $\lambda$.

## 6 Related Works

### 6.1 Machine Translation Quality Estimation

Most supervised QE methods transfer knowledge using the pre-training and fine-tuning strategy. Predictor-Estimator (Kim et al., 2017) and QE-Brain (Fan et al., 2019) pre-train the QE model with the word prediction task. The translation language modeling objective (Conneau and Lample, 2019) is also used for QE pre-training in (Kepler et al., 2019). As mentioned in (Cui et al., 2021), these pre-training objectives are far from the QE task, and the pseudo data method achieves remarkable results on several QE datasets. COMET (Rei et al., 2020) and UniTE (Wan et al., 2022) incorporated the QE task with MT evaluation.

A series of unsupervised QE works use various features to estimate the adequacy and fluency of MT outputs. Etchegoyhen et al. (2018) use lexical translation overlaps and language model cross-entropy scores. BERTScore (Zhang et al., 2019), YiSi (Lo, 2019) and Zhou et al. (2020) adopt token similarities based on contextual embeddings. Fomicheva et al. (2020) define several QE features based on translation probability, uncertainty quantification, and attention scores. Furthermore, SSQE (Zheng et al., 2021) utilizes TLM generation probabilities of masked MT words and explores Monte Carlo Dropout (Gal and Ghahramani, 2016) for quantifying uncertainty. On the other hand, pseudo data can be directly used for unsupervised QE. However, SSQE outperforms existing pseudo data methods due to noisy pseudo data. In this paper, we aim to improve the quality of pseudo data.

### 6.2 Constrained Machine Translation

There are two main ways to integrate constraints in NMT: constrained training and decoding. Song et al. (2019) and Dinu et al. (2019) train a variant MT model by augmenting the data to include the constraints. Constraint-aware beam search algorithms have been widely used to introduce constraints during inference (Hokamp and Liu, 2017; Hu et al., 2019). The above approaches rely on pre-defined constraints, while our work also defines desired constraints to generate better pseudo data. The closest work is (Lopes et al., 2019), which penalizes all tokens not present in translations with the same reduction so that the automatic post-editing model will not over-correct the translations. For comparison, we assign different rewards to reference tokens based on their positions and generation probabilities.

## 7 Conclusion

We present a constrained beam search algorithm to generate improved pseudo QE data. The proposed CBSQE preserves the main structure of references by increasing the generation probabilities of reference tokens according to their original NMT probabilities and positions. Experiments show that we achieve remarkable performance in both supervised and unsupervised settings. The analysis further shows that CBSQE is efficient, and each part of CBSQE contributes to the improvement.

## Limitations

In this study, we only explore the pseudo data methods for the word- and sentence-level QE tasks with post-editing annotation. The proposed CBS algorithm could be effective in other evaluation tasks. For instance, CBS can be extended to other QE annotations with simple designs as introduced in (Geng et al., 2022). CBS could also be extended to the fully unsupervised QE task by using back-translations of an unsupervised MT as pseudo parallel data; Some annotations for the translation metric are also based on fine-grained errors (Freitag et al., 2022), making it natural to extend CBS to reference-based evaluation; Zhou et al. (2021) developed a token-level hallucination detection method similar to BSQE, which can be further improved by the CBS algorithm; High-quality pseudo data could be also helpful to predict human feedback. Predicting human feedback is crucial for aligning language models with human intent (Ouyang et al., 2022).

The labels of some CBSQE pseudo data are still noisy. It could be further improved by developing powerful fine-grained reference-based evaluation methods or employing the self-distillation strategy. We have not tested the ability of CBSQE to generate various pseudo data using the same parallel pair. Besides, there is a potential risk of collapse under adversarial attacks for QE methods. We leave these explorations in the future.

## Acknowledgements

We would like to thank the anonymous reviewers for their insightful comments. Shujian Huang is the corresponding author. This work is supported by National Science Foundation of China (No. 62376116, 62176120), the Liaoning Provincial Research Foundation for Basic Research (No. 2022-KF-26-02).

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

# A  Other Implementation Details

All experiments set the random seed as 1 except for fine-tuning. When fine-tuning, we run each experiment with different random seeds five times. We follow the setting of the WMT shared tasks to report the best results in Table 2. We also report average results in Table 7. We can observe similar phenomena to the main results from average results. For SSQE, we follow the setting of

| Dataset | Settings | Method | Sent-level Test | Word-level Test | | | |
|---------|----------|--------|-----------------|-----------------|---|---|---|
| | | | Pearson↑ | MCC↑ | F1-MULT↑ | F1-OK↑ | F1-BAD↑ |
| WMT19 EN-DE | Supervised | XLMRQE | 52.06 | 38.75 | 42.64 | 92.16 | 46.29 |
| | | DirectQE | 55.95 | 42.26 | 45.87 | **92.77** | 49.45 |
| | | BSQE | 56.16 | 41.63 | 45.67 | 91.56 | 49.88 |
| | | **CBSQE** | **56.50** | **42.60** | **46.45** | 92.15 | **50.41** |
| WMT20 EN-ZH | Supervised | XLMRQE | 59.38 | 44.76 | 50.63 | 73.30 | 69.08 |
| | | DirectQE | 63.52 | 48.99 | 54.62 | 74.61 | 73.21 |
| | | BSQE | 64.44 | 48.71 | 54.82 | 74.32 | 73.76 |
| | | **CBSQE** | **65.29** | **49.44** | **55.22** | **74.81** | **73.81** |
| WMT21 EN-ZH | Supervised | XLMRQE | 28.92 | 27.84 | 35.46 | **79.74** | 44.47 |
| | | DirectQE | 31.41 | 29.86 | 36.07 | 78.52 | 45.93 |
| | | BSQE | 31.14 | 29.36 | 34.87 | 76.65 | 45.49 |
| | | **CBSQE** | **31.59** | **30.15** | **36.41** | 78.91 | **46.13** |

Table 7: Average results for supervised methods on different QE datasets.

hyper-parameters described in (Zheng et al., 2021). We provide code for pre-processing data and all source code in supplementary materials. We list the hyper-parameters as follows.

## A.1 Generating Pseudo Data

### A.1.1 DirectQE

For the generator, we follow the setting in DirectQE, and one NVIDIA V100 GPU is used to train the model. The mask ratio is 30% for the WMT19 EN-DE direction and 35% for WMT20/21 EN-ZH direction, respectively. We use the Adam optimizer with $\beta_1 = 0.9$, $\beta_2 = 0.999$ to optimize model parameters. The initial learning rate is set to 0.2, and the Noam learning rate schedule is equipped with 8000 warm-up steps. We update the parameters every 20 batches, and the maximum number of tokens is set to 8000 in a batch. If the validation performance gets no improvements for the last 20 runs, we will perform early stopping.

### A.1.2 BSQE and CBSQE

For WMT19 EN-DE, we train a translation model with one NVIDIA V100 GPU. Adam optimizer parameters $\beta_1$ and $\beta_2$ are set to 0.9 and 0.98, respectively, to optimize model parameters, and the weight decay is set to 1e-4. We set the initial learning rate to 5e-4 and use the inverse square root learning schedule with 4000 warm-up steps. The dropout rate is set to 0.3, and the penalty of label smoothing is set to 0.1. During training, we set the maximum number of tokens in a batch to 4096 and set the update frequency to 1. We perform early stopping if the validation performance does not improve for the last 15 runs. For WMT20/21 EN-ZH, we use the translation model provided by

WMT20/21.

For CBSQE Pseudo QE data, we manually tune hyper-parameters one by one. We test several groups of hyper-parameters (temperature=5; lambda=0.15, 0.2; epsilon=0.25, 0.3, 0.35). Finally, we set the temperature to 5. The $\lambda$ is set to 0.15 for WMT19 EN-DE and 0.20 for WMT20/21 EN-ZH. The $\epsilon$ is set to 0.35 for WMT19 EN-DE and 0.30 for WMT20/21 EN-ZH. For self-ensemble, we predict the QE labels 40 times which is the same as SSQE.

During data generation, we set the beam size to 4 and set the batch size to 512 for both BS and CBS.

## A.2 Pre-Training and Fine-tuning

For unsupervised experiments, three RTX 3090Ti GPUs are used to train the models. We set the learning rate to 1e-5 and use the Adam optimizer with $\beta_1 = 0.9$, $\beta_2 = 0.999$ to optimize model parameters. The clip norm is set to 1.0. During training, we set the maximum number of tokens in a batch to 1600 tokens, and the update frequency is set to 3. We perform 5 epochs for every experiment and choose the best valid checkpoint.

For supervised experiments, one NVIDIA V100 GPU is used to train the models. We use the Adam optimizer with $\beta_1 = 0.9$, $\beta_2 = 0.999$ to optimize model parameters, and the clip norm is set to 1.0. The learning rate is set to 2e-5 for the WMT19 EN-DE direction, 5e-6 for the WMT20 EN-ZH direction, and 1e-5 for the WMT21 EN-ZH direction. During training, we set the maximum number of sentences in a batch to 5, and the update frequency is set to 20. We perform early stopping if the validation performance does not improve for the last 5

| Dataset | Settings | Method | Sent-level Test | | |
|---|---|---|---|---|---|
| | | | Pearson↑ | MAE↓ | RMSE↓ |
| WMT19 EN-DE | Unsupervised | DirectQE | 42.52 | 12.88 | 20.53 |
| | | BSQE | 46.30 | 23.46 | 27.78 |
| | | SSQE (w/ dropout) | 45.04 | 668.56 | 668.78 |
| | | **CBSQE** | 48.47 | 20.93 | 25.30 |
| | Supervised | XLMRQE | 52.21 | 11.27 | 17.10 |
| | | DirectQE | 56.26 | 11.49 | 18.25 |
| | | BSQE | 56.45 | 11.34 | 17.12 |
| | | **CBSQE** | 56.65 | 11.06 | 17.04 |
| WMT20 EN-ZH | Unsupervised | DirectQE | 45.15 | 34.59 | 39.00 |
| | | BSQE | 54.99 | 14.33 | 17.85 |
| | | SSQE (w/ dropout) | 43.82 | 676.87 | 677.14 |
| | | **CBSQE** | 59.09 | 13.66 | 17.07 |
| | Supervised | XLMRQE | 60.55 | 13.88 | 17.34 |
| | | DirectQE | 63.59 | 12.97 | 16.40 |
| | | BSQE | 64.71 | 12.95 | 16.27 |
| | | **CBSQE** | 65.42 | 12.85 | 16.16 |
| WMT21 EN-ZH | Unsupervised | DirectQE | 20.35 | 20.82 | 26.03 |
| | | BSQE | 21.22 | 43.41 | 49.28 |
| | | SSQE (w/ dropout) | 21.40 | 641.26 | 641.83 |
| | | **CBSQE** | 27.41 | 43.35 | 49.71 |
| | Supervised | XLMRQE | 30.38 | 23.07 | 27.05 |
| | | DirectQE | 31.99 | 23.30 | 28.22 |
| | | BSQE | 31.88 | 27.44 | 32.07 |
| | | **CBSQE** | 32.15 | 24.24 | 28.99 |

Table 8: Main results of MAE and RMSE metrics on different QE test sets.

epochs for the WMT19 EN-DE direction and the WMT20 EN-ZH direction and 10 epochs for the WMT21 EN-ZH direction.

# B Results of MAE and RMSE Metrics

Table 8 shows the results of MAE and RMSE metrics, CBSQE achieves the best MAEs and RMSEs in many cases. Since SSQE is not trained to minimize MAE loss, its MAEs and RMSEs are abnormally high. As shown in (Graham, 2015), the MAE metric has a counter-intuitive effect on QE system rankings. Therefore, we provide these results only for reference.

# C Significance Test

As recommended in (Specia et al., 2021), we use William's test[4] to compute statistical significance on Pearson. The results are listed in Table 9 and 10.

---

[4]https://github.com/ygraham/mt-qe-eval

| Dataset | Settings | Method | DirectQE | BSQE | SSQE (w/ dropout) | **CBSQE** |
|---|---|---|---|---|---|---|
| WMT19 EN-DE | Unsupervised | DirectQE | - | - | - | - |
| | | BSQE | 0.1024 | - | - | - |
| | | SSQE (w/ dropout) | 0.1551 | 0.2878 | - | - |
| | | **CBSQE** | **0.0128** | **0.0076** | **0.0454** | - |
| WMT20 EN-ZH | Unsupervised | DirectQE | - | - | - | - |
| | | BSQE | **2.54E-6** | - | - | - |
| | | SSQE (w/ dropout) | 0.2782 | **3.29E-6** | - | - |
| | | **CBSQE** | **1.85E-10** | **0.0007** | **1.69E-11** | - |
| WMT21 EN-ZH | Unsupervised | DirectQE | - | - | - | - |
| | | BSQE | 0.3745 | - | - | - |
| | | SSQE (w/ dropout) | 0.3306 | 0.4455 | - | - |
| | | **CBSQE** | **0.0074** | **0.0001** | **0.0140** | - |

Table 9: Results of William's test for unsupervised methods. The $p$-values are marked in bold if $p < 0.05$.

| Dataset | Settings | Method | XLMRQE | DirectQE | BSQE | **CBSQE** |
|---|---|---|---|---|---|---|
| WMT19 EN-DE | Supervised | XLMRQE | - | - | - | - |
| | | DirectQE | **0.0049** | - | - | - |
| | | BSQE | **0.0041** | 0.4978 | - | - |
| | | **CBSQE** | **0.0067** | 0.4716 | 0.4634 | - |
| WMT20 EN-ZH | Supervised | XLMRQE | - | - | - | - |
| | | DirectQE | **0.0492** | - | - | - |
| | | BSQE | **0.0108** | 0.1810 | - | - |
| | | **CBSQE** | **0.0005** | **0.0266** | 0.0990 | - |
| WMT21 EN-ZH | Supervised | XLMRQE | - | - | - | - |
| | | DirectQE | 0.3104 | - | - | - |
| | | BSQE | 0.1892 | 0.3328 | - | - |
| | | **CBSQE** | 0.1405 | 0.2581 | 0.4340 | - |

Table 10: Results of William's test for supervised methods. The $p$-values are marked in bold if $p < 0.05$.