# OpenReview forum: "Improved Pseudo Data for Machine Translation Quality Estimation with Constrained Beam Search"
_EMNLP/2023/Conference — EMNLP 2023 Main_

### Official Review · Reviewer_Ehgu · 2023-07-28

**Typos Grammar Style And Presentation Improvements:** The description of 3.1 is a bit confu…
**Soundness:** 3

**Excitement:**

3: Ambivalent: It has merits (e.g., it reports state-of-the-art results, the idea is nice), but there are key weaknesses (e.g., it describes incremental work), and it can significantly benefit from another round of revision. However, I won't object to accepting it if my co-reviewers champion it.

**Paper Topic And Main Contributions:**

This paper proposes a new data-augmentation method for translation quality estimation (QE). The author performs constrained beam search on the MT model and manipulate the generation process according to MT probabilities. They make comparison with other data-augmentation methods in both supervised and zero-shot scenarios to verify their effectiveness.

**Reasons To Accept:**

1. The proposed data-augmentation method only manipulate the decoding process, without demanding the post-train process.
2. It is enlightening to alleviating the deviance by constrained beam search.
3. Results on both supervised and zero-shot setting are improved significantly compared with previous methods based on the same data.

**Reasons To Reject:**

1. The proposed method relies on a pre-trained MT model to create augmentation data, which is not applicable for a lot of cases (such low-resource languages).
2 The calculation of HTER label relied on automatic scripts can hardly be scaled to other annotations such as MQM or DA. If it is not the case, then the author should experiment on other languages and annotations to verify the scalability.
3. The contribution is focused on the constrained beam search for QE, which is somewhat lesss qualified for a long paper.

**Reproducibility:**

4: Could mostly reproduce the results, but there may be some variation because of sample variance or minor variations in their interpretation of the protocol or method.

**Reviewer Confidence:**

4: Quite sure. I tried to check the important points carefully. It's unlikely, though conceivable, that I missed something that should affect my ratings.

---

> ### Author Rebuttal · Authors · 2023-08-29
>
> Thank you for your time and the positive evaluation of the soundness. We would like to discuss the following concerns in the final paper.
>
> > R1: The proposed method relies on a pre-trained MT model to create augmentation data, which is not applicable for a lot of cases (such as low-resource languages).
>
> In many applications, we have access to the target MT model that needs to be evaluated. Following this trend, WMT QE shared tasks have adopted the white-box setting in recent years [1]. In the black-box setting, we can obtain a competitive multilingual MT model [2] for low-resource languages.
>
> The parallel data is the bottleneck of pseudo data methods for low-resource languages. As shown in Figure 3, CBSQE still achieves competitive results with 100K parallel pairs which could satisfy many language pairs. We leave the exploration of fully unsupervised pseudo data methods in the future. Because it is quite heavy to include both supervised, unsupervised, and fully unsupervised settings in one paper. Considering that DirectQE and SSQE only focus on the supervised and unsupervised setting respectively.
>
> > R2: The calculation of HTER label relied on automatic scripts can hardly be scaled to other annotations such as MQM or DA. If it is not the case, then the author should experiment on other languages and annotations to verify the scalability.
>
> As shown in the WMT2022 QE shared task paper [3], the advanced pseudo data methods for MQM (rank 1st on EN-DE sentence- and word-level, rank 2nd on Multilingual sentence-level) and DA (rank 2nd on English-Marathi word-level) obtain pseudo labels based on simple rules. For the MQM annotation, they sample tokens from top-2, 10, and 100 NMT probabilities to simulate minor, major, and critical errors. Then, the MQM scores can be calculated as the weighted sum of different errors, which is the definition of MQM. For the DA annotation, they just take the replaced ratio of reference tokens as the DA score.
>
>  Although different annotations are proposed for different applications, they all reflect the quality of translations and share similar word-level tags. The analysis of the pseudo data quality (section 5.1) is based on word-level labels and NMT probabilities which are not dependent on the specific annotation. Hence, CBSQE generates improved pseudo translations and word-level labels for different annotations. The sentence scores of different annotations can be obtained with simple designs as introduced above.
>
>
> > R3: The contribution is focused on the constrained beam search for QE, which is somewhat less qualified for a long paper.
>
> This paper highlights that better pseudo data concurrently need MT-like pseudo translations and accurate quality labels. To this end, it is essential to define the proper constraints of beam search. CBSQE needs to decide when to improve the probability of references and which reference tokens should be improved. CBSQE only improves the probability of references when the probability of any reference token is above a threshold so that the less fluent hypothesis will not be mistakenly rewarded. CBSQE uses a distance function to assign larger improvements for closer reference tokens from the right side so that we can preserve the main structure of references and therefore obtain accurate labels using TER tools. As shown in section 5.1, CBSQE successfully improves pseudo data quality. Ablation studies further show that each part of the constraints is important for improvement.
>
> > Writing: The description of 3.1 is a bit confusing and could be improved.
>
> We will polish the expressions of the method part and revise the paper carefully.
>
>
> [1] https://www.statmt.org/wmt21/quality-estimation-task.html
>
> [2] Tang et al. Multilingual Translation with Extensible Multilingual Pretraining and Finetuning. arXiv:2008.00401.
>
> [3] Geng et al. NJUNLP’s Participation for the WMT2022 Quality Estimation Shared Task. WMT2022.

---

### Official Review · Reviewer_Zcy9 · 2023-08-04

**Soundness:** 3

**Excitement:**

3: Ambivalent: It has merits (e.g., it reports state-of-the-art results, the idea is nice), but there are key weaknesses (e.g., it describes incremental work), and it can significantly benefit from another round of revision. However, I won't object to accepting it if my co-reviewers champion it.

**Paper Topic And Main Contributions:**

This paper addresses the task of MT quality estimation.
The authors propose to generate pseudo data using MT model with constrained beam search.
The proposed method (CBSQE) keeps the main structure of references by increasing the generation probabilities of the reference tokens with regards to to their original translation probabilities and positions.
Experiments were conducted on the WMT19 (English-German)  shared task and the WMT20 and WMT21 (English-Chinese) shared tasks in both supervised and unsupervised scenarios.
The proposed method (CBSQE) was compared to several related work baselines: DirectQE, BSQE, the unsupervised method  SSQE in addition to a standard supervised baseline XLMRQE.
The results overall show improvements in most cases by the proposed method in both supervised and unsupervised scenarios.

**Questions For The Authors:**

Q1: As quality estimation measures at the sentence level, in addition to Pearson correlation factor,  I would expect the mean absolute error (MAE) and the root-mean-square deviation (RMSE) as two other informative metrics usually used. Can the authors provide the results for these two measures?
Q2: Did the authors run a significance test on the obtained results? Such as T-test for instance?


**Reasons To Accept:**

- Overall the paper is clear and well written
- The proposed method shows improvements over the baselines

**Reasons To Reject:**

- In many cases the improvements are not very important
- Missing important metrics at the sentence level evaluation (MAE and RMSE)
- It is not clear to me if the results are significant

**Reproducibility:**

2: Would be hard pressed to reproduce the results. The contribution depends on data that are simply not available outside the author's institution or consortium; not enough details are provided.

**Reviewer Confidence:**

3: Pretty sure, but there's a chance I missed something. Although I have a good feel for this area in general, I did not carefully check the paper's details, e.g., the math, experimental design, or novelty.

**Typos Grammar Style And Presentation Improvements:**

- line 275 "...based on" instead of "...base on"

---

> ### Author Rebuttal · Authors · 2023-08-29
>
> We would like to thank you for the constructive suggestions.
> >R1: In many cases the improvements are not very important.
>
> Yes, for all supervised scenarios, the improvements seem incremental. As discussed in the paper, QE models tend to learn shortcuts (such as lexical artifacts, and fluency) from real QE datasets [1]. As a result,  it seems that the benefits of improved pseudo data are moderate after fine-tuning.
>
> >R2: Missing important metrics at the sentence level evaluation (MAE and RMSE).
>
> >Q1: As quality estimation measures at the sentence level, in addition to the Pearson correlation factor, I would expect the mean absolute error (MAE) and the root-mean-square deviation (RMSE) as two other informative metrics usually used. Can the authors provide the results for these two measures?
>
> As shown in [2], the MAE metric has a counterintuitive effect on QE system rankings. Therefore, we didn't report MAE and RMSE metrics in the main results as many QE works do (e.g., SSQE, and the official WMT QE shared task paper[3]). We would like to add MAE and RMSE metrics to the final version for more information.
>
> >R3: It is not clear to me if the results are significant
>
> >Q2: Did the authors run a significance test on the obtained results? Such as T-test for instance?
>
> We have performed the significance test on unsupervised results.
> As recommended in [3], we use [William’s test](https://github.com/ygraham/mt-qe-eval) to compute statistical significance on Pearson.
> The results are listed as follows, and the p-values are marked in bold if p < 0.05.
> CBSQE significantly outperforms baselines on all datasets.
>
> |  |  | WMT19 EN-DE |  |  |
> | :----: | :----: | :----: | :----: | :----: |
> |  | **DirectQE** | **BSQE** | **SSQE** | **CBSQE** |
> | **DirectQE** | - | - | - | - |
> | **BSQE** | 0.1024 | - | - | - |
> | **SSQE** | 0.1551 | 0.2878 | - | - |
> | **CBSQE** | **0.0128** | **0.0076** | **0.0454** | - |
>
> |  |  | WMT20 EN-ZH |  |  |
> | :----: | :----: | :----: | :----: | :----: |
> |  | **DirectQE** | **BSQE** | **SSQE** | **CBSQE** |
> | **DirectQE** | - | - | - | - |
> | **BSQE** | **2.54E-6** | - | - | - |
> | **SSQE** | 0.2782 | **3.29E-06** | - | - |
> | **CBSQE** | **1.85E-10** | **0.0007** | **1.69E-11** | - |
>
> |  |  | WMT21 EN-ZH |  |  |
> | :----: | :----: | :----: | :----: | :----: |
> |  | **DirectQE** | **BSQE** | **SSQE** | **CBSQE** |
> | **DirectQE** | - | - | - | - |
> | **BSQE** | 0.3745 | - | - | - |
> | **SSQE** | 0.3306 | 0.4455 | - | - |
> | **CBSQE** | **0.0074** | **0.0001** | **0.0140** | - |
>
> We have not performed the significance test on supervised results due to the limited time. We would like to provide these results in the final paper. However, we suppose that CBSQE could not significantly outperform baselines in the supervised setting. The reason is the same as we discussed for R1.
>
> [1] Sun et al. Are we Estimating or Guesstimating Translation Quality? ACL2020.
>
> [2] Yvette Graham. Improving Evaluation of Machine Translation Quality Estimation. ACL 2015.
>
> [3] Specia et al. Findings of the WMT 2021 Shared Task on Quality Estimation. WMT2021.

---

### Official Review · Reviewer_JQ3q · 2023-08-11

**Soundness:** 3

**Excitement:**

3: Ambivalent: It has merits (e.g., it reports state-of-the-art results, the idea is nice), but there are key weaknesses (e.g., it describes incremental work), and it can significantly benefit from another round of revision. However, I won't object to accepting it if my co-reviewers champion it.

**Paper Topic And Main Contributions:**

The paper proposed a reranking approach to do quality estimation for Machine Translation output. A separate model is being trained to do quality estimation on the fly during the decoding process(beam search) and rank higher quality output higher in the beam.

**Reasons To Accept:**

Paper is well formed with lots of experiments and have enough details of the experiment setups. It also covers different aspect like efficiency and abdication study.

**Reasons To Reject:**

I don't have any reason to reject this paper

**Reproducibility:**

4: Could mostly reproduce the results, but there may be some variation because of sample variance or minor variations in their interpretation of the protocol or method.

**Reviewer Confidence:**

1: Not my area, or paper was hard for me to understand. My evaluation is just an educated guess.

---

> ### Author Rebuttal · Authors · 2023-08-29
>
> Thank you for your time and positive comments. We would like to highlight the contribution as follows. Pseudo data methods play an essential role in the QE area due to the lack of labeled QE data. However, existing pseudo data methods suffer from either noisy pseudo transaltions or labels. In this paper, we proposed a constrained beam search algorithm (CBSQE) that reconciles the pseudo label quality with pseudo translation quality. Analysis shows that CBSQE improves pseudo data quality. Therefore CBSQE outperforms baselines in both supervised and unsupervised settings.

---

### Official Review · Reviewer_tS9S · 2023-08-11

**Soundness:** 3

**Ethical Concerns:**

Yes

**Excitement:**

3: Ambivalent: It has merits (e.g., it reports state-of-the-art results, the idea is nice), but there are key weaknesses (e.g., it describes incremental work), and it can significantly benefit from another round of revision. However, I won't object to accepting it if my co-reviewers champion it.

**Paper Topic And Main Contributions:**

This paper target the "false negative" problem of pseudo data construction for quality estimation, i.e., the previous methods may ignore semantic equivalence and produce pseudo data that are labeled as "BAD" but actually they are reasonable translation. They propose an MT-based model equipped with a constrained beam search to alleviate this problem. The motivation makes sense to me. The experiments show that the method works better than baselines in both supervised and unsupervised settings, and the analysis shows that the proposed model results in better pseudo labels.

**Reasons To Accept:**

1. Better performances compared with the baselines.
2. The main idea is simple and straightforward.

**Reasons To Reject:**

The writing is unclear: I suggest that the author proofread and modify the article's expression, especially the causal logic and the method part. I often have to guess the meaning of some sentences: E.g., in lines 194-198, "Thus, we assign the same probabilities to ... To this end, we define the distance function ...". This kind of portion is rather challenging for me to comprehend as I progress through it.

On the whole, I value the core concept of the method, which is straightforward and yields consistent improvements in both supervised and unsupervised scenarios. Nonetheless, owing to certain instances of unclear articulation, such as in the method section (although I can grasp the general gist), I hold reservations regarding the method's practicality and ease of adoption within the community.






**Reproducibility:**

3: Could reproduce the results with some difficulty. The settings of parameters are underspecified or subjectively determined; the training/evaluation data are not widely available.

**Reviewer Confidence:**

4: Quite sure. I tried to check the important points carefully. It's unlikely, though conceivable, that I missed something that should affect my ratings.

**Typos Grammar Style And Presentation Improvements:**

line 174-176: I can roughly understand your meaning, but I suggest saying it straightforwardly in case some misleading.

The expression of the method part, including the Algorithm1, needs to polish.

---

> ### Author Rebuttal · Authors · 2023-08-29
>
> We would like to thank you for your time and valuable suggestions for writing.
>
> > R1: The writing is unclear...Nonetheless, owing to certain instances of unclear articulation, such as in the method section (although I can grasp the general gist), I hold reservations regarding the method's practicality and ease of adoption within the community.
>
> We will polish the expressions of the method part and revise the paper carefully. We would like to clear the practicality as follows. Generally speaking, we improve the generation probabilities of some reference tokens during beam search. The adjustments happen only if the probabilities of any reference token are above a threshold so that the less fluent hypothesis will not be mistakenly rewarded. We want to preserve the main structure of references so that we can obtain accurate labels using TER tools. Therefore, CBSQE assigns larger improvements for closer reference tokens when adjusting. As shown in section 5.1, CBSQE successfully improves the pseudo data quality. Ablation studies further show that each part of the constraints is important for improvement. The whole algorithm is efficient (section 5.3) and open source (supplementary materials).

---

### Meta-Review · Area_Chair_Pbb1 · 2023-09-22

**Recommendation:** 3

**Metareview:**

The authors propose a method to create synthetic data for training an MT QE model using a machine translation (MT) model and constrained beam search. The main idea is simple (which is good!), straightforward and leads to quality improvements. The author might want to improve the writing at some points in the paper. The reviewers overall agree that this is solid work and suggested only minor changes.

---

### Decision · Program_Chairs · 2023-10-07

**Decision:**

Accept-Main

**Comment:**

The authors propose a method to create synthetic data for training an MT QE model using a machine translation (MT) model and constrained beam search. The main idea is simple (which is good!), straightforward and leads to quality improvements. The author might want to improve the writing at some points in the paper. The reviewers overall agree that this is solid work and suggested only minor changes.